# Effects of Two Different Straw Pellets on Yak Growth Performance and Ruminal Microbiota during Cold Season

**DOI:** 10.3390/ani13030335

**Published:** 2023-01-17

**Authors:** Xiangyan Wang, Bingang Shi, Zhi Zuo, Youpeng Qi, Shijie Zhao, Xueping Zhang, Lijuan Lan, Yu Shi, Xiu Liu, Shaobin Li, Jiqing Wang, Jiang Hu

**Affiliations:** Gansu Key Laboratory of Herbivorous Animal Biotechnology, College of Animal Science and Technology, Gansu Agricultural University, Lanzhou 730070, China

**Keywords:** yaks, rumen microbiota, straw pellets feed, growth performance, volatile fatty acids, cold season

## Abstract

**Simple Summary:**

The Tianzhu white yaks is a unique species living on the Qinghai–Tibet Plateau and has strong adaptability to high altitude and low oxygen. These yaks mainly rely on natural pasture as the source of nutrients. However, because of low temperature and insufficient pasture in winter, the Tianzhu white yak lose weight. The ruminal microbiota of yaks can efficiently acidify and hydrolyze fiber to produce volatile fatty acids (VFAs). In this study, we investigated how the ruminal microbiota affects the growth performance of yaks after supplementing straw feed and assessed the changes in ruminal microbiota and VFAs. Correlation analysis revealed a correlation among the ruminal microbiota, VFAs, and daily weight gain (ADG) of yaks. Mainly, the abundance of cellulose bacteria that can hydrolyze straw feed increased to produce more levels of VFAs, which were absorbed by the yaks, increasing their ADG.

**Abstract:**

The Tianzhu white yaks (*Bos grunniens*) live on the Qinghai–Tibet Plateau. During winter, a lack of resources and low nutritional levels seriously affect their growth performance. In this study, we aimed to explore the effect of supplementation straw pellets on the growth performance and ruminal microbiota of yaks. Overall, at 6 (6M, *n* = 24), 18 (18M, *n* = 26), 30 (30M, *n* = 20), 42 (42M, *n* = 24), and 54 (54M, *n* = 22) month old Tianzhu white yaks were selected (total *n* = 116) and divided into the mixed straw + grazing (MSG), corn straw + grazing (CSG), and the grazing control (G) groups according to age and gender. Their growth performance was measured as per different dietary treatments. The rumen microbial community structure and levels of VFAs were analyzed from the 6M, 30M, and 54M male yaks from each group. The supplementary diets led to an increase in the ADG, which was the highest in the MSG group. The MSG group exhibited the highest level of acetate and total VFAs (TVFAs) among the three groups (*p* < 0.05). In addition, the 16S rRNA sequencing results proved that the microbial composition was dominated by the members of *Firmicutes* and *Bacteroidetes*. *Christensenellaceae R-7 group* was significantly abundant in the CSG and MSG groups compared to the G group (*p* < 0.05). Principal coordinate analysis (PCoA) revealed that the bacterial community structure of rumen in the MSG and CSG groups was considerably different from that in the G group; 6M samples exhibited different rumen microbial diversity compared with the other samples. Correlation analysis revealed that *Christensenellaceae_R-7_group* was positively correlated with the levels of acetate, TVFAs, and ADG. These results demonstrated that mixed straw pellets improved the growth performance of yaks, increased the abundance of *Christensenellaceae R-7_group* involved in cellulose degradation in the rumen, and produced large amounts of VFAs, which were absorbed by yaks, thus increasing their ADG. This study provides new insights into the effects of straw pellet supplementation on the changes in the rumen microbiota and growth performance of yaks.

## 1. Introduction

The Tianzhu white yak is a unique species living on the Qinghai–Tibet Plateau 2500 m above sea level. These yaks have strong adaptability to high altitude and low oxygen. They are known as “the pearl of the plateau”, as they are rare semi-wild animals in China and the world [1]. Only approximately 100,000 Tianzhu white yaks are present in Tianzhu Tibetan Autonomous County, accounting for approximately 7% of the total number of yaks in the world [2]. The Tianzhu white yaks play an important role in the economy and agriculture of the Qinghai–Tibet Plateau by providing meat, milk, and wool to local herdsmen [3]. Compared with other beef, the yak meat has fine texture, high protein content, low content of fat, and rich levels of amino acids, making it more popular among locals [4]. However, the slow growth rate and low meat production of yaks seriously affect their economic value. Improving yak growth to increase meat production has become the focus of yak breeding. Under traditional management, the Tianzhu white yaks rely on native pasture to meet their nutritional needs, and their weight varies due to seasonal fluctuations in feed supply [5]. During the cold season, low temperature and deficiency of pasture result in reduced nutrition, slow growth, less weight, and even death. To improve their growth performance, it is especially important to combine grazing with supplementary feeding.

The rumen is a natural fermentation system, inhabited mainly by bacteria, fungi, protozoa, and other microorganisms, which are responsible for degrading the feed that enters the rumen [6]. Ruminants rely on the ruminal microbiota to degrade proteins, dietary fiber, and starch and to convert them into VFAs, which are absorbed by the ruminants [7]. In addition, ruminal microbes play an important role in nutrient absorption and intestinal health in ruminants [8]. The rumen microbiota in yaks can use lignocellulose from crop straw to produce microbial proteins and fatty acids with high nutritional value, which are important in the formation of yak products such as meat, milk, and wool. At present, the effect of diet on the rumen microbiota in yaks is a popular research area. Significant differences in the yak ruminal microbiota were reported during different grazing periods [9]. The supplementation of concentrates may change the core ruminal microbiota group in yaks [10]. Similarly, great differences in microbial diversity were observed between grazing yaks and yaks with feed supplementation [11]. Moreover, the structural composition of the yak ruminal microbiota changed after supplementation with a 3:7 ratio of concentrate to roughage [12]. Although the above studies illustrated that the ruminal microbiota in the yaks changes under different conditions, it is unclear how the ruminal microbiota affects the growth performance of yaks.

Lignocellulosic biomass is the most abundant renewable resource in the world [13]. Crop straw is one of the most abundant lignocellulosic materials. However, the vast majority of crop straw is burned directly in the field, which leads to not only environmental pollution, but also wastage of the resource [14]. The rumen of ruminants is a natural fermentation system, with complex and stable microorganisms, which can effectively degrade and acidify crop straw to produce VFAs [15]. In recent years, oat grass, corn straw, and alfalfa hay have been widely used by pastoralists during the cold season. These materials provide energy and supplement minerals, fatty acids, microorganisms, and proteins, thus reducing weight loss in ruminants [16]. Therefore, more crop straw is used as pellet feed for production. Forage pellet size and physical change affect rumen digestion, digestion rate, and rumen protein synthesis, and rumen protein synthesis is increased by 15% when long grass is replaced with ground hay in cow diets [17]. Compared with the total mixed ration diet, addition of straw feed to the diet increased the daily weight gain of goat [18]. Michał Gąsiorek et al. reported that alfalfa hay supplementation in feed diet helped improve the pH of the bovine rumen [19]. Furthermore, the addition of straw feed could increase the unsaturated fatty acid content in lamb meat [20] and could increase dry matter intake, promoting calf growth [21]. Therefore, supplemental feeding of crop straw to yaks in winter can improve not only the rumen environment, but also their growth performance.

A previous study reported that species, diet, age, and season affect ruminal microbiota, with diet being a key factor [22]. The aim of this study was to investigate the ruminal microbiota of yaks in response to straw supplements during the cold season. We hypothesized that straw supplements would alter the bacterial composition of yaks; the products of rumen fermentation would be altered, thereby affecting the growth of yaks. This study can improve our understanding of variations in the rumen microbiota in yaks in response to changes in diet and provide a theoretical basis for the use of crop straw to improve the production performance of yaks.

## 2. Materials and Methods

All experimental designs and feeding management involving animals were approved by the Animal Ethics Committee of Gansu Agricultural University (Approval number 2006-398). The feeding experiment was conducted at the herdsman farm (Tianzhu, Gansu Province, China).

### 2.1. Animals, Experimental Design, and Feeding

A total of 116 Tianzhu white yaks were selected from Tianzhu Country, Gansu Province, China (altitude: 2500 m; average temperature: −9 to 6 °C). They were of various ages: 6 (6M, *n* = 24; 13 males, 11 females), 18 (18M, *n* = 26; 15 males, 11 females), 30 (30M, *n* = 20; 10 males, 10 females), 42 (42M, *n* = 24; 9 males, 15 females), and 54 (54M, *n* = 22; 10 males, 12 females) months old. These yaks were randomly divided into three groups of diet treatments on the basis of gender and age, with at least three male and three female yaks of each age. The three groups were mixed straw + grazing (MSG), corn straw + grazing (CSG), and grazing control (G). Mixed straw consisted of 50% corn straw, 20% oat grass, 20% alfalfa grass, 9% corn, and 1% premix additives (Appendix A). The nutrient composition of feeds of the MSG and CSG groups is shown in Table 1. These yaks grazed in the same pasture with free access to water between 7:00 a.m. and 6:00 p.m. The pastures in the area were characterized by alpine meadows, and the main vegetation was *Kobresia graminifolia*, *K. humilis*, *Elymus nutan*, *K. pygmaea*, and *Melissitus ruthenica*. During the supplementary feeding period, each 6M, 18M, 30M, 42M, and 54M yak in the MSG and CSG groups was fed with 1, 2, 3, 4, and 4.5 kg of mixed straw pellets and corn straw pellets, respectively, at 6:00 p.m. every day. The total feeding period was 54 days including an adaptation period of 7 days. After the supplementation experiment, three male yaks (6M, 30M, and 54M) from each of the three groups (a total of 27 male yaks) were slaughtered. 

### 2.2. Assessment Growth Performance 

Yaks were weighed before the morning grazing during the supplementary feeding period. The ADG of yaks was calculated according to the following formula: ADG (kg/day) = (FBW − IBW)/t. Here, IBW, FBW, and t represent the initial body weight, final body weight, and feeding days, respectively.

### 2.3. Sample Collection

The mixed straw and corn straw pellets (3–5 cm) were made using a straw cutter. Corn straw and mixed straw pellets were collected and used to determine the nutrient content.

After the supplementary feeding period ended, three male yaks (6M, 30M, and 54M) from each group (a total of 27 male yaks) were selected and slaughtered after fasting for 24 h and being deprived of water for 8–12 h. The fluids in the rumen were collected. The detailed procedure is as follows: the entire rumen was collected, and the ruminal contents were scooped up using a sterile scalpel, filtered through four layers of sterile gauze, and divided into two 50 mL sterile cryogenic tubes. The tubes were labeled and immediately stored in liquid nitrogen for subsequent 16S rRNA gene analysis and determination of VFAs.

### 2.4. Feed Analysis

Corn straw and mixed straw was dried in an oven at 60 °C for 48 h until a constant weight. They were ground with a grinder, and the nutrient content was determined by 1 mm screening. The acid detergent fiber and neutral detergent fiber concentrations in corn straw pellets and mixed straw pellets samples were determined using the Van Soest method [23]. Dry matter and crude protein were measured using AOAC methods [24]. 

### 2.5. Determination of Rumen Fermentation Parameters

The frozen samples were thawed at 4 °C and centrifuged at 3000× *g* at 4 °C for 10 min. The VFAs were measured using gas chromatography (Agilent 6890N, Santa Clara, CA, USA) with a chromatographic column (HP 19091N-213), according to the method of Zhang et al. [25]. Briefly, the column nitrogen flow rate was controlled at 46.3 cm/s, the sampling amount was 0.4 µL, and the inlet temperature was maintained at 220 °C. The initial column temperature was set at 110 °C and held for 30 s. Further, the temperature was increased at 10 °C/min to 120 °C and maintained for 4 min. Finally, the temperature was increased at 10 °C/min to 150 °C. The detector temperature was maintained at 250 °C.

### 2.6. DNA Extraction and Analysis of Bacterial Community in Rumen

Microbial DNA was extracted from ruminal fluids using the EZNAsoil DNA kit (Invitek, Germany). DNA quality was checked with 1% agarose gel electrophoresis. The DNA was diluted to 1 ng/μL with ultrapure water for subsequent processing. The V3-V4 region of bacterial 16S rRNA genes was amplified using universal primers 341F: CCTACGGGNGGCWGCAG and 806R: GGACTACHVGGGTATCTAAT [26]. PCR reaction conditions were as follows: 3 min of predenaturation at 95 °C; denaturation at 95 °C for 30 s, annealing at 55 °C for 30 s, and extension at 72 °C for 45 s (a total of 27 cycles); final extension at 72 °C for 10 min. The amplicons were purified using AxyPrep DNA Gel Extraction Kit (Axygen Biosciences, Union City, CA, USA). All amplicons were sequenced and analyzed on an Illumina HiSeq (2500 PE250) platform (Illumina, San Diego, CA, USA).

### 2.7. Bioinformatic Analysis

On the basis of the raw data obtained from the Illumina HiSeq 2500 PE250 sequencing platform (Illumina, San Diego, CA, USA), original sequence quality was controlled and spliced using the Fastp software perform (https://ccb.jhu.edu/software/FLASH/index.shtml, accessed on 28 May 2022) [27]. The sequences were clustered to operational taxonomic units (OTUs) using UPARSE software performs (http://drive5.com/uparse/, accessed on 28 May 2022) based on 97% similarity [28]. The Uchime algorithm was used to eliminate chimeras by Usearch software. Each sequence was classified and annotated at different levels (phylum and genus) through the Ribosomal Database Project (RDP) [29]. The alignment database (SSU123) was compared with the entries in the Silva database [30]. Alpha diversity (Shannon and Simpson) and richness (Ace and Chao) were analyzed using the mother software platform (https://www.mothur.org/wiki/Downloadmothur, accessed on 28 May 2022) [31]. Beta diversity analysis was performed to compare differences in species diversity (microbial composition and structure) between different samples using Qiime (http://qiime.org/install/index.html, accessed on 28 May 2022). The Kruskal–Wallis H-test method was used to identify differences between groups. The linear discriminant analysis (LDA) effect size (LEfSe) method was used to evaluate the differences in the microbial communities, and the LDA score threshold was 4.

### 2.8. Statistical Analyses

Correlation analysis among VFAs, ADG, and the rumen microbiota was performed using R software (version 3.3.1) (Rcore team, Vienna, Austria). One-way ANOVA in SPSS software (version 26.0) (SPSS Inc., Chicago, IL, USA) was used to analyze the yak growth performance and VFA levels. The data were represented as the means ± SEMs. A *p*-value ≤0.05 was considered significant.

## 3. Results

### 3.1. Growth Performance of Yaks

The effects of straw pellet supplementation on the growth performance of yaks of different ages are shown in Table 2. After 47 days of supplementation, the MSG group exhibited higher body weight than the G group (*p* < 0.05). The ADG was significantly higher in the MSG groups than in the G group. It was significantly higher in the CSG groups than that in the G group in 30M and 42M yaks (*p* < 0.05). The ADG was higher in the MSG group than in the CSG group in 6M and 42M yaks (*p* < 0.05). The ADG increased significantly with age in the MSG and CSG groups (Table 3). No significant effect of gender was observed on the ADG (Table 4). The interaction of age and diets had a significant effect on the ADG of yaks, but the interaction of age, gender, and diets had no significant effect on the ADG of yaks (Table 5).

### 3.2. Parameters of Rumen Fermentation

The levels of VFAs in the ruminal fluids of yaks are given in Table 6 and Table 7. Supplemental feeding affected some VFAs in yaks. The levels of TVFAs and acetate, and the acetate/propionate ratio (A/P) were significantly higher (*p* < 0.05), while the propionate level was significantly lower (*p* < 0.05) in the MSG group than in the CSG and G groups in 6M yaks; in addition, isobutyrate and valerate levels were lower in the MSG and CSG groups than in the G group (*p* < 0.05). The level of acetate was significantly higher, while the butyrate and valerate levels were lower in the MSG and CSG groups than in the G group (*p* < 0.05) in 30M yaks. The levels of acetate and TVFAs were higher, while the valerate level was lower in the MSG group than in the G group in 54M yaks (*p* < 0.05). In the CSG group, the levels of butyrate and isovalerate at 6M were greater than at 30M, while the acetate level was lower than at 30M (*p* < 0.05).

### 3.3. Sequencing of the Ruminal Microbiota

A total of 300,443,472 PE reads were obtained in this study, and 1,402,461 clean reads were generated through quality control and filtering of the raw data, with an average of 51,943 sequences per sample. The effective sequences of all samples were clustered in 2520 OTUs with 97% similarity, including 1547, 1595, 1589, 1647, 1753, 1464, 1641, 1545, and 1653 OTUs in MSG6, CSG6, G6, MSG30, CSG30, G30, MSG54, CSG54, and G54, respectively, and 16, 20, 24, 12, 11, 49, 20, 50, and 23 unique OTUs in MSG54, CSG54, G54, MSG6, CSG6, G6, MSG30, CSG30, and G30, respectively (Figure 1).

### 3.4. Alpha Diversity of Rumen Microbiota

For 30M yaks, alpha diversity results indicated higher values of ACE and Chao indices in the MSG and CSG groups than in the G group (Table 7, *p* < 0.05). Alpha diversity was not different among various ages in the same dietary treatment group (Table 8 and Table 9).

### 3.5. Beta Diversity of Rumen Microbiota

The MSG and CSG groups exhibited similar rumen microbiota diversity, which was clearly different from that of the G group (Figure 2A–C), which indicated that supplementary diets affected the structure of microbial communities. The 30M and 54M yaks had similar rumen microbial diversity, while the rumen microbial diversity of 6M yaks was clearly different from that of 30M and 54M yaks, indicating that calves and adults formed different rumen microbiota (Figure 2D–F).

### 3.6. The Variation in the Rumen Microbiota

At the phylum level, 25 phyla were detected across all groups. The relative abundance of the main bacterial phyla is shown in Figure 3. The total relative abundance of *Firmicutes* and *Bacteroidete* accounted for >91%; they were the dominant bacteria in all groups. Compared with the G group, the relative abundance of *Firmicutes* increased while that of *Bacteroidetes* decreased in the MSG and CSG groups. However, the difference was not significant among groups (*p* > 0.05).

At the genus level, 351 genera were detected. Among them 84 genera had a relative abundance >0.1%. The relative abundance of bacterial genera is shown in Figure 4. *Christensenellaceae_R-7_group*, *Rikenellaceae_RC9_gut_group*, and *Prevotella* were the top three genera of ruminal microbiota in all groups. The relative abundance of *Christensenellaceae_R-7_group*, *lachnospiraceae_NK3A20_group* and *NK4A214_group* in the MSG and CSG groups was significantly higher than that in the G group (*p* < 0.05) in 6M yaks (Figure 5A), The relative abundance of *Christensenellaceae R-7 group* was higher (*p* < 0.05) in the MSG and CSG groups than that in the G group in 30M yaks (Figure 5B). Moreover, the abundance of some genera significantly changed among different age groups under the same feeding conditions. In the CSG group (Figure 5C), *lachnospiraceae_NK3A20_group* was more abundant in 54M yaks than in 6M and 30M yaks (*p* < 0.05). The relative abundance of *Christensenellaceae_R-7_group* was significantly higher in 54M and 6M yaks than in 30M yaks (*p* < 0.05) in the G group (Figure 5D). LEfSe analysis revealed that *Peptostreptococcales-Tissierellale*, and *Anaerovoracaceae* were significantly enriched in the MSG 54M yaks, while genera *Christensenellaceae_R-7_group* and *Prevotella* were significantly enriched in the CSG 30M and G 30M yaks, respectively (Figure 6).

### 3.7. Correlations of Microbial Communities with VFAs and ADG

The correlations between bacterial genera in the rumen and fermentation parameters are shown in Figure 7. The ADG was positively correlated with relative abundance of *Christensenellaceae_R-7_group*, *norank_f__UCG-011*, and *Lachnospiraceae_NK3A20_group*. The acetate levels were positively correlated with the relative abundance of *Christensenellaceae_R-7_group* and *NK4A214_group*, but negatively correlated with *Ruminococcus*. The TVFA levels were positively correlated with *NK4A214_group*, but negatively correlated with *Rikenellaceae_RC9_gut_group*. The butyrate levels were negatively correlated with *Succiniclasticum* and *Christensenellaceae_R-7_group*. The valerate and propionate levels were negatively correlated with *Christensenellaceae_R-7_group* and *NK4A214_group*.

## 4. Discussion

The climate characteristics of the Qinghai–Tibet Plateau region determine the seasonal imbalance of productivity of natural grassland and forage nutrient output. Each year is divided into warm and cold seasons, and the vegetation cover of the grassland corresponds to the grazing and dry grass periods [32]. The quantity and quality of available pasture is low during the dry period. This makes the grazing yaks lose weight and exhibit the growth trend of “thin in winter and dead in spring”. In our study, the ADG of yaks in the G group exhibited a decreasing trend; however, the MSG and CSG groups exhibited a positive effect on the ADG of yaks during the cold grazing season. The decrease in pasture quantity during the cold season may be the main reason for weight loss, and the low temperature increases the maintenance needs. Thus, the nutrients obtained from the pasture cannot meet the maintenance needs, resulting in weight loss in yaks. Compared with the G group, the ADG of yaks increased in the MSG and CSG groups. This may be attributed to increased nutrient intake and faster digestion rate in the MSG and CSG groups, leading to improved growth performance and feed conversion efficiency. Dietary energy utilization is the main factor influencing the performance of livestock, and the growth efficiency of livestock mainly depends on the energy level of the diet. The MSG group exhibited better growth performance than the CSG group, which can be attributed to the higher energy in the MSG group than in the CSG group. This result is consistent with a previous study which reported that yaks supplemented with high- to medium-energy diets exhibited better growth performance than those supplemented with low-energy diets [33]. The yaks in the MSG group received more energy than those in the CSG group, which may be the reason why the growth performance of the MSG group was better than that of the CSG group. VFAs are the main product of carbohydrate digestion by rumen microbiota and are the main source of energy for hosts; they are absorbed into the rumen epithelium [34,35]. A/P is related to the energy utilization efficiency; a low A/P indicates high energy efficiency of feed [32]. In the present study, A/P of the MSG group decreased with age. Our results suggest that the energy utilization efficiency of yaks increased with age, which explains the more pronounced improvement in growth performance with increasing age of yaks in this study. The energy utilization efficiency observed in this study is consistent with a previous study in which adult cattle used energy from the diet more efficiently than calves to a certain degree [36]. Microbial abundance and diversity could change the propionic acid level, such as the abundance of *Propionibacterium* and *Acidipropionibacterium* correlated with the propionate level [37]. In our study, we observed a significant decrease in propionate levels in the MSG group compared with those in the G group. Similarly, correlation analysis revealed that *Christensenellaceae_R7_group* and *Ruminococcaceae_NK4A214_group* were negatively correlated with propionate level. Therefore, the decrease in propionate level of the MSG group may be due to microbial composition and abundance. VFAs are the main fermentation products of the ruminal microbiota; thus, molar ratios of individual VFAs in the rumen may predict changes in the composition of the rumen microflora [38]. We believe that a supplementary diet will alter not only the available substrates for fermentation, but also the rumen bacterial community composition, which will affect the growth performance of ruminants. Therefore, changes in bacterial communities in the yak rumen might be one of the reasons for the changes in the levels of VFAs.

Yaks can live and reproduce on the Tibetan plateau, which is related to the population of the ruminal microbiota [39]. A previous study reported that dietary fiber is a major source of energy for intestinal bacterial fermentation and can shape the intestinal flora [40]. Moreover, fiber intake increases the levels of dietary microorganism diversity [41]. In this study, we sequenced the rumen microbiota of yaks fed on various straw feeds and observed that the abundance of the microbiota was significantly different among the three groups. In this study, *Bacteroides* and *Firmicutes* were the dominant bacteria in the rumen of yaks, consistent with a previous study [42]. *Firmicutes* are mainly responsible for increasing the number of bacterial genes encoding enzymes that help the host in digesting cellulosic material and absorbing nutrients [43,44]. *Bacteroides* are responsible for hydrolyzing carbohydrates and proteins [45,46]. The ratio of *Firmicutes* and *Bacteroides* (F/B) is used as an important parameter to assess the impact of microorganisms on the energy requirements of host [47]. A higher F/B can help host in absorbing energy-related nutrients and maintaining body balance more effectively [48]. On the other hand, increased F/B is positively related to the growth rate and feed efficiency of cattle [49]. In this study, the F/B of the CSG and MSG groups was higher than that of the G group. Therefore, supplementary straw feeds could help yaks in absorbing energy efficiently, keeping the body in balance, and increasing the growth performance. This further explained the significant increase in ADG of yaks in the CSG and MSG groups. This was consistent with a previous study, which reported that supplementation with high-fiber diets increased the F/B [50].

The dominant bacterial genera were *Rikenellaceae_RC9_gut_group*, *Christensenellaceae_R-7_group*, and *Prevotella* in all groups, consistent with previous studies [51,52]. At the bacterial genus level, some differences were observed in different groups and ages of yaks. *Lachnospiraceae_NK3A20_group* and *Christensenellaceae_R-7_group* are beneficial microbial communities and were the dominant bacterial genera observed in this study. *Christensenellaceae_R-7_group* belongs to Christensenellaceae family, which can produce α-arabinosidase, β-glucosidase, and β-galactosidase associated with feed conversion efficiency [53]. *Christensenellaceae_R-7_group* contains genes for cellulase and hemicellulase secretase, which improve the ability of ruminants to degrade cellulose and obtain energy from polysaccharides that are difficult to degrade [54] and positively correlate with weight [55]. In the present study, we observed that the relative abundance of *Christensenellaceae_R-7_group* increased after dietary supplementation and positively correlated with ADG and acetate level. This indicated that *Christensenellaceae_R-7_group* could help to degrade and acidify cellulose and hemicellulose in straw feed. We inferred that supplementary diet altered the levels of VFAs by affecting the relative abundance of *Christensenellaceae_R-7_group*, thereby improving growth performance of yaks. Therefore, *Christensenellaceae_R-7_group* had a positive effect on maintaining the energy balance and body weight of yaks during the cold season. *Lachnospiraceae* belongs to *Firmicutes*; they are involved in degrading cellulose and hemicellulose in plants and converting them into VFAs, which are absorbed by host and used as energy [56,57,58]. We observed that the relative abundance of *lachnospiraceae_NK3A20_group* was the highest in the MSG group in 6M yaks, which could partly explain the highest levels of VFAs in the MSG group in 6M yaks. In addition, *lachnospiraceae_NK3A20_group* is mainly present in the gastrointestinal tract and plays a role in maintaining its health. In 6M yaks, the rumen and flora were immature, and extra feed restricted calves’ digestion and absorption of feed. Thus, *lachnospiraceae_NK3A20_group* participated in the positive regulation of the rumen environment related to intestinal inflammation and release of cytotoxic factors. *Lachnospiraceae_NK3A20_group* is associated with immune regulation and healthy homeostasis to help calves in adapting to harsh living environment and reducing the incidence of intestinal diseases [59]. A previous study reported a significant positive correlation between *Ruminococcaceae_NK4A214_group* and glycolysis [60]. Acetyl-CoA produced by acetate can be preferentially used in the synthesis of citric acid [61]. In this study, the abundance of *Ruminococcaceae_NK4A214_group* increased significantly after supplementation of straw feed, and correlation analysis revealed that the abundance of *Ruminococcaceae_NK4A214_group* significantly and positively correlated with acetate level. Therefore, we hypothesize that *Ruminococcaceae_NK4A214_group* may influence acetate production through the glycolytic pathway, further leading to the difference in total VFAs between supplemented (MSG and CSG) and grazing (G) groups of yaks. Therefore, yaks in the MSG and CSG groups could absorb more VFAs through the rumen epithelium, resulting in higher growth performance of yaks. In addition, *Ruminococcus* is known as a cellulolytic bacterial species that produces various enzymes to degrade cellulose into acetate [62]; indeed, it negatively correlated with acetate level in this study. This may be due to differences in species. A higher abundance of *Ruminococcus* was reported in the rumen of natural grazing yaks [63]. Similar results were reported in the rumen of steers fed with high-fiber feed [64], consistent with their importance in the rumen of the G group in the present study. Furthermore, *Prevotella* is known as the most abundant genus of bacteria in the rumen; its main function is to participate in the processing of complex dietary polysaccharides and to promote the uptake and absorption of monosaccharides by host [65]. Moreover, it plays an important role in the degradation of cellulose and vegetable proteins [43]. *Prevotella* participates in the degradation of cellulose, production of VFAs, and breakdown of plant polysaccharides into monosaccharides, thus supplying energy to yaks. This is the reason behind its higher relative abundance in the G group because of increased fiber and lignin contents, as well as low nutrient content in the alpine meadow. Consequently, yaks must increase their feed intake to fight cold weather during winter.

Correlation results revealed an association between the rumen microbiota and VFAs. *Rikenellaceae_RC9_gut_group* is present in the rumen of yaks and common cattle and is closely related to carbohydrate and nitrogen utilization [51,66]. *Rikenellaceae_RC9_gut_group* is mainly responsible for the degradation of plant-based polysaccharides. The abundance of acetate and propionate-producing microorganisms increased when the relative abundance of *Rikenellaceae_RC9_gut_group* decreased [67]. We observed that the relative abundance of *Rikenellaceae_RC9_gut_group* negatively correlated with the total VFAs. This indicated that mixed straw pellet feed changed the VFA levels by affecting the abundance of *Rikenellaceae_RC9_gut_group* and improved the growth performance of yaks. A previous study reported that *Christensenellaceae_R-7_group* was positively correlated with acetate level [68]. Acetyl-CoA can utilize acetate to synthesize citric acid [61]. In this study, the *Christensenellaceae_R-7_group* was positively correlated with acetate level and ADG. Therefore, we inferred that the relative abundance of *Christensenellaceae_R-7_group* may affect the level of acetate via glycolysis pathway, further influencing the ADG of yaks.

## 5. Conclusions

In summary, this study revealed that the growth performance and TVFA levels in the MSG and CSG groups were significantly higher than those in the G group. Compared with the CSG group, the MSG group exhibited better growth performance. The relative abundance of cellulolytic bacteria, such as *Christensenellaceae_R-7_group*, which contributed to the degradation and fermentation of lignocellulose, was higher in the MSG and CSG groups than in the G group. Correlation analysis revealed that the relative abundance of *Christensenellaceae_R-7_group* positively correlated with ADG and acetate level. *Christensenellaceae_R-7_group* might have a certain influence on acetate level through glycolysis, and it ultimately improved the growth performance of yaks. The VFA levels and abundance of cellulosic flora varied with age during the same feeding treatment. Thus, different dietary compositions and ages affected fermentation and bacterial diversity in the rumen of yaks, and supplementing mixed straw pellet feed could help yaks to better adapt to the special environment of the Qinghai–Tibet plateau.

## Figures and Tables

**Figure 1 animals-13-00335-f001:**
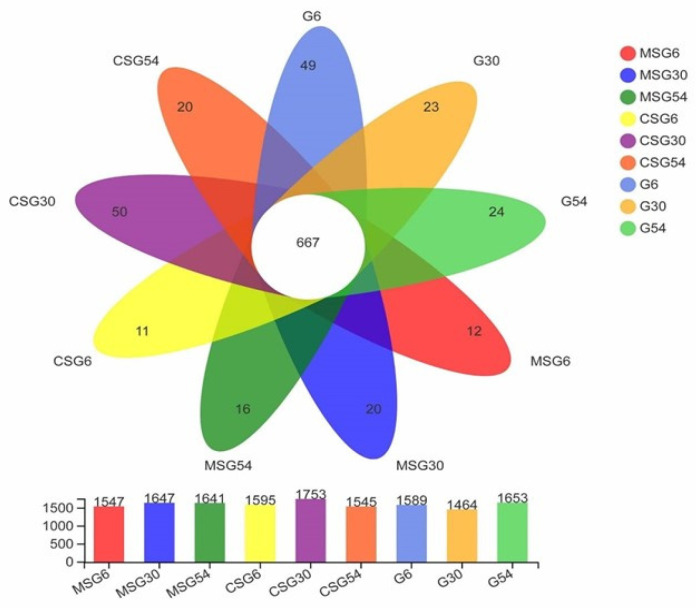
Venn diagram of luminous bacterial OTUs. Parts of no overlap represent unique OTUs of diets at 6M, 30M, and 54M. MSG represents composite straw pellets and grazing; CSG represents corn straw pellets and grazing; G represents grazing.

**Figure 2 animals-13-00335-f002:**
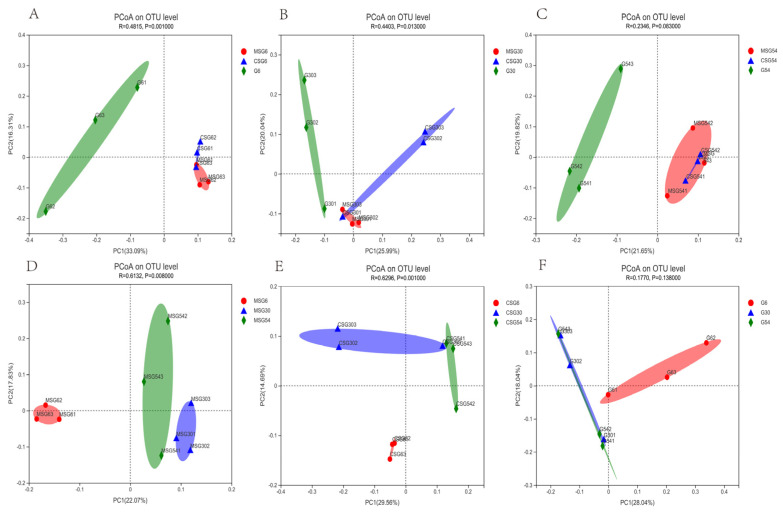
PCoA plots using Bray−Curtis dissimilarity based on ASVs of rumen microbiota in yaks. (**A**–**C**) The effect of supplementation of straw pellets on the beta diversity of rumen microbiota in yaks. (**D**–**F**) The effect of age on the beta diversity of rumen microbiota in yaks.

**Figure 3 animals-13-00335-f003:**
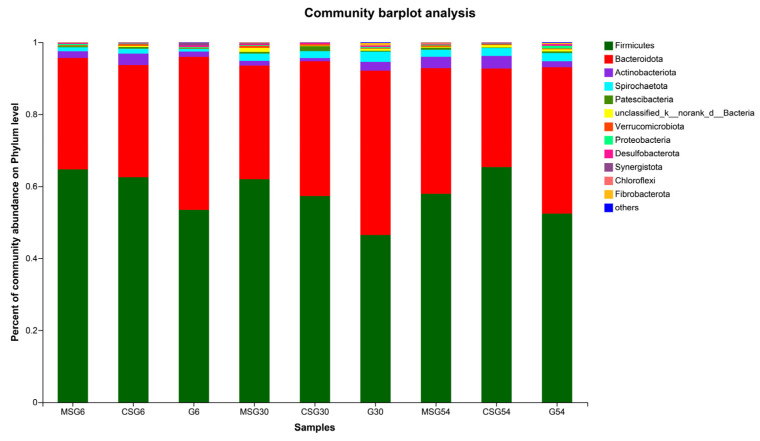
Microbial composition in the rumen of yaks at the phylum level.

**Figure 4 animals-13-00335-f004:**
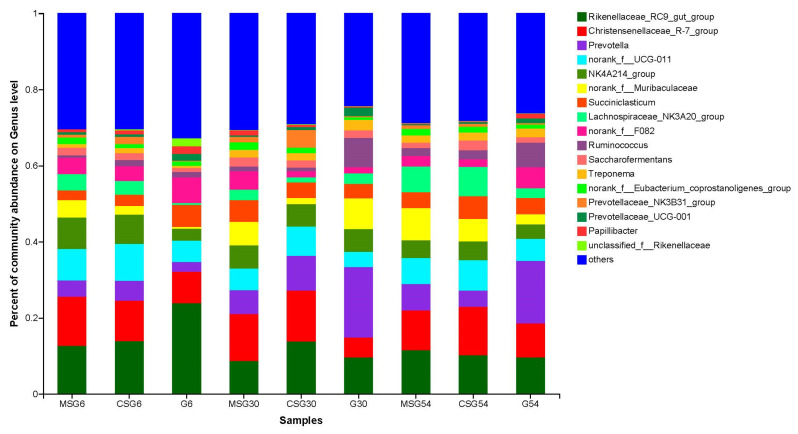
Microbial composition in the rumen of yaks at the genera level.

**Figure 5 animals-13-00335-f005:**
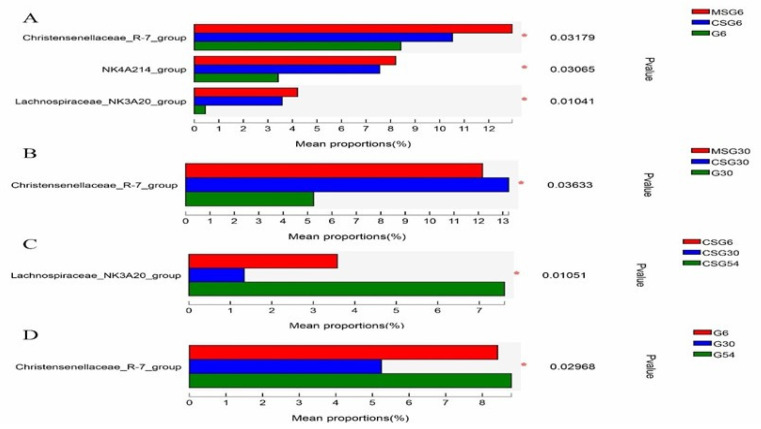
Difference in the relative abundance of various genera according to one-way ANOVA test; * *p* < 0.05. A: Comparison of dominant genera in 6M with different dietary treatments; B: Comparison of dominant genera with different dietary treatments in 30M; C: Comparison of dominant genera with different ages in the CSG group; D: Comparison of dominant genera with the different ages in the G group.

**Figure 6 animals-13-00335-f006:**
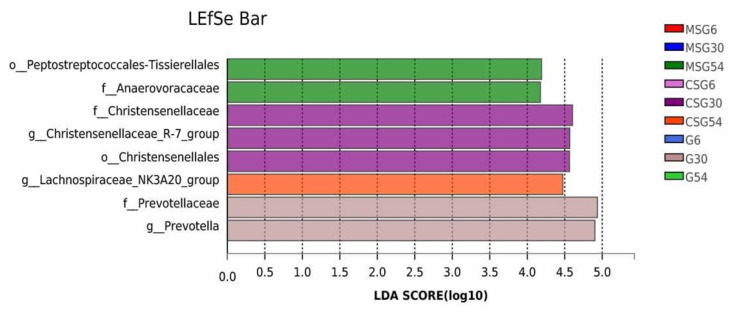
LEfSe results in different ages and supplements. Prefixes represent abbreviations for the taxonomic rank of each taxon; (f-) represents the identification of bacterial taxa significantly different from each group at the family level by LEfSe, (g-) represents the identification of bacterial taxa significantly different from each group at the genus level by LEfSe, and (o-) represents the identification of bacterial taxa significantly different from each group at the order level by LEfSe.

**Figure 7 animals-13-00335-f007:**
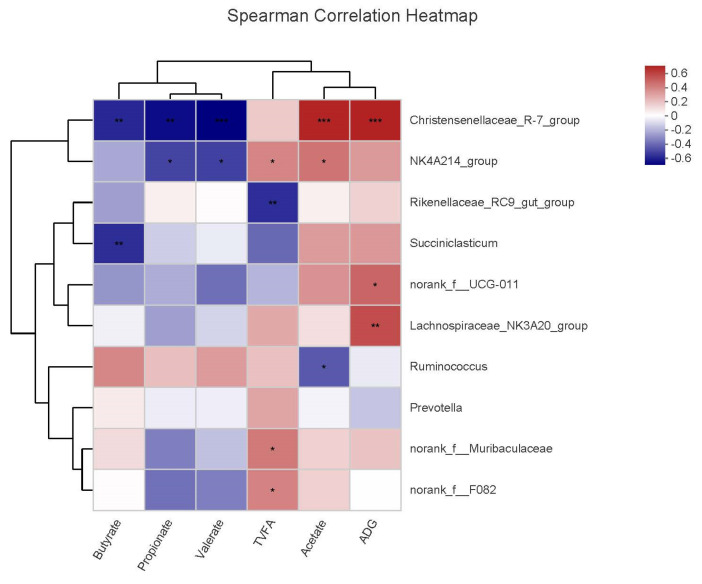
Heat map of the correlation among the relative abundance of microorganisms, VFAs, and ADG. * *p* < 0.05, ** *p* < 0.01, *** *p* < 0.001.

**Table 1 animals-13-00335-t001:** Nutrient composition of roughage (after air drying).

Diet Treatment	Dry Matter (%)	Crude Protein (%)	Neutral Detergent Fiber (%)	Acid Detergent Fiber (%)
Corn straw	91.57	3.30	63.55	41.98
Mixed straw	94.76	7.09	58.78	38.79

**Table 2 animals-13-00335-t002:** Body weight and average daily gain of Tianzhu white yaks supplemented with various diets.

Month	Gender	Number	Growth Properties	Group			
				MSG	CSG	G	*p*
6M	Male	N = 5 (MSG)	IBW (kg)	51.90 ± 2.76	48.50 ± 3.76	45.00 ± 1.22	0.207
		N = 4 (CSG)	FBW (kg)	59.60 ± 1.72 ^a^	51.50 ± 2.73 ^b^	44.95 ± 1.64 ^c^	0.001
		N = 4 (G)	ADG (kg/d)	0.13 ± 0.02 ^a^	0.05 ± 0.01 ^b^	0.00 ± 0.01 ^b^	0.005
	Female	N = 4 (MSG)	IBW (kg)	45.87 ± 3.47	42.66 ± 1.33	45.62 ± 1.66	0.628
		N = 3 (CSG)	FBW (kg)	54.37 ± 4.98 ^a^	45.16 ± 0.16 ^ab^	40.75 ± 0.75 ^b^	0.038
		N = 4 (G)	ADG (kg/d)	0.09 ± 0.02 ^a^	0.03 ± 0.01 ^b^	−0.01 ± 0.00 ^c^	0.002
18M	Male	N = 5 (MSG)	IBW (kg)	109.26 ± 2.05	106.83 ± 0.44	106.50 ± 1.05	0.369
		N = 3 (CSG)	FBW (kg)	122.80 ± 1.39 ^a^	113.00 ± 2.51 ^b^	104.28 ± 1.89 ^c^	<0.001
		N = 7 (G)	ADG (kg/d)	0.15 ± 0.02 ^a^	0.07 ± 0.02 ^a^	−0.02 ± 0.02 ^b^	0.001
	Female	N = 3 (MSG)	IBW (kg) FBW (kg) ADG (kg/d)	111.50 ± 6.33 125.33 ± 6.88 ^a^ 0.15 ± 0.03 ^a^	108.83 ± 3.63 114.66 ± 2.18 ^ab^ 0.06 ± 0.02 ^b^	104.20 ± 1.71 106.20 ± 2.76 ^b^ −0.02 ± 0.02 ^b^	0.367 0.010 0.004
		N = 3 (CSG)					
		N = 5 (G)					
30M	Male	N = 4 (MSG)	IBW (kg)	153.25 ± 8.95	144.0 ± 7.11	124.62 ± 8.21	0.092
		N = 3 (CSG)	FBW (kg)	165.00 ± 7.93 ^a^	150.00 ± 8.24 ^a^	122.87 ± 8.25 ^b^	0.015
		N = 3 (G)	ADG (kg/d)	0.21 ± 0.03 ^a^	0.14 ± 0.02 ^a^	−0.03 ± 0.01 ^b^	0.001
	Female	N = 4 (MSG)	IBW (kg) FBW (kg) ADG (kg/d)	159.87 ± 9.64 177.50 ± 7.59 ^a^ 0.20 ± 0.03 ^a^	147.50 ± 7.64 158.66 ± 7.31 ^ab^ 0.12 ± 0.04 ^a^	134.00 ± 10.88 131.00 ± 10.54 ^b^ −0.03 ± 0.01 ^b^	0.193 0.015 0.002
		N = 3 (CSG)					
		N = 3 (G)					
42M	Male	N = 3 (MSG)	IBW (kg)	203.16 ± 2.08	202.66 ± 4.60	198.33 ± 4.25	0.638
		N = 3 (CSG)	FBW (kg)	229.66 ± 0.66 ^a^	216.66 ± 4.05 ^b^	197.33 ± 4.80 ^c^	0.002
		N = 3 (G)	ADG (kg/d)	0.30 ± 0.03 ^a^	0.16 ± 0.02 ^b^	−0.01 ± 0.00 ^c^	<0.001
	Female	N = 4 (MSG)	IBW (kg) FBW (kg) ADG (kg/d)	198.75 ± 12.19 218.75 ± 13.40 ^a^ 0.22 ± 0.01 ^a^	192.00 ± 3.04 205.33 ± 2.40 ^ab^ 0.15 ± 0.05 ^b^	191.87 ± 2.70 190.37 ± 2.85 ^b^ −0.01 ± 0.00 ^c^	0.701 0.026 <0.001
		N = 3 (CSG)					
		N = 8 (G)					
54M	Male	N = 4 (MSG)	IBW (kg)	239.62 ± 7.08	231.66 ± 3.00	221.83 ± 1.42	0.125
		N = 3 (CSG)	FBW (kg)	259.75 ± 9.47 ^a^	243.16 ± 1.01 ^ab^	221.33 ± 1.76 ^a^	0.015
		N = 3 (G)	ADG (kg/d)	0.36 ± 0.08 ^a^	0.23 ± 0.07 ^ab^	−0.01 ± 0.01 ^b^	0.026
	Female	N = 4 (MSG)	IBW (kg)	219.87 ± 2.97 238.00 ± 7.57 ^a^ 0.20 ± 0.06 ^a^	212.50 ± 3.43 222.50 ± 3.79 ^ab^ 0.11 ± 0.01 ^ab^	207.50 ± 9.18 206.25 ± 8.87 ^b^ 0.01 ± 0.02 ^b^	0.372 0.034 0.025
		N = 4 (CSG)	FBW (kg)				
		N = 4 (G)	ADG (kg/d)				

Note: IBW = initial body weight, FBW = final body weight, ADG = average daily gain, MSG = mixed straw + grazing, CSG = corn straw + grazing, and G = grazing. 6M, 18M, 30M, 42M, and 54M represent 6, 18, 30, 42, and 54-month old yaks, respectively. The same applies to the tables below. Superscripts a, b, and c in the same row indicate a significant difference (*p* < 0.05) in different dietary supplements at the same age.

**Table 3 animals-13-00335-t003:** The average daily gain of Tianzhu white yaks various with ages.

Diet Treatment	Growth Properties	Gender	Month					
			6M	18M	30M	42M	54M	*p*
MSG	ADG (kg/day)	Male (*n* = 21)	0.13 ± 0.12 ^c^	0.15 ± 0.02 ^bc^	0.21 ± 0.03 ^bc^	0.30 ± 0.03 ^ab^	0.36 ± 0.08 ^a^	0.013
		Female (*n* = 19)	0.09 ± 0.02	0.15 ± 0.03	0.20 ± 0.03	0.22 ± 0.01	0.20 ± 0.06	0.166
CSG	ADG (kg/day)	Male (*n* = 16)	0.05 ± 0.01 ^b^	0.07 ± 0.02 ^b^	0.14 ± 0.02 ^ab^	0.16 ± 0.02 ^ab^	0.23 ± 0.07 ^a^	0.024
		Female (*n* = 16)	0.03 ± 0.01	0.06 ± 0.02	0.12 ± 0.04	0.15 ± 0.05	0.11 ± 0.01	0.131
G	ADG (kg/day)	Male (*n* = 20)	0.00 ± 0.01	−0.02 ± 0.02	−0.03 ± 0.02	−0.01 ± 0.00	−0.01 ± 0.01	0.862
		Female (*n* = 24)	−0.01 ± 0.00	−0.02 ± 0.02	−0.04 ± 0.02	−0.01 ± 0.00	0.01 ± 0.02	0.294

Note: Superscripts a, b, and c in the same row indicate a significant difference (*p* < 0.05) in different ages with the same dietary supplements and gender.

**Table 4 animals-13-00335-t004:** The average daily gain of Tianzhu white yaks various with gender.

Growth Properties	Diet Treatment	Gender	Month				
			6M	18M	30M	42M	54M
MSG	ADG (kg/day)	Male (*n* = 21)	0.13 ± 0.02	0.15 ± 0.02	0.21 ± 0.03	0.30 ± 0.03	0.36 ± 0.08
		Female (*n* = 19)	0.09 ± 0.02	0.15 ± 0.03	0.20 ± 0.03	0.22 ± 0.01	0.20 ± 0.06
		*p*	0.432	0.948	0.072	0.072	0.204
CSG	ADG (kg/day)	Male (*n* = 16)	0.05 ± 0.01	0.07 ± 0.02	0.14 ± 0.02	0.16 ± 0.02	0.23 ± 0.07
		Female (*n* = 16)	0.03 ± 0.01	0.06 ± 0.02	0.12 ± 0.04	0.15 ± 0.05	0.11 ± 0.01
		*p*	0.274	0.845	0.818	0.868	0.115
G	ADG (kg/day)	Male (*n* = 20)	0.00 ± 0.01 ^a^	−0.02 ± 0.02	−0.03 ± 0.02	−0.01 ± 0.00	−0.01 ± 0.01
		Female (*n* = 24)	−0.01 ± 0.00 ^b^	−0.02 ± 0.02	−0.04 ± 0.02	−0.01 ± 0.00	0.01 ± 0.02
		*p*	0.420	0.852	0.860	0.353	0.503

Note: Superscripts a and b in the same column indicate a significant difference (*p* < 0.05) in different genders with the same dietary supplements and age.

**Table 5 animals-13-00335-t005:** Effect of different supplemental feeding on average daily gain of yaks of different genders and ages.

Item		ADG
Gender	male	0.11 ± 0.01
	female	0.07 ± 0.01
Age	6M	0.04 ± 0.01 ^a^
	18M	0.05 ± 0.01 ^a^
	30M	0.10 ± 0.02 ^ab^
	42M	0.11 ± 0.02 ^ab^
	54M	0.15 ± 0.03 ^b^
Diets	MSG	0.20 ± 0.01^a^
	CSG	0.11 ± 0.01 ^b^
	G	−0.01 ± 0.01 ^c^
*p*		
Gender		0.129
Age		0.010
Diets		<0.001
Age × gender		0.252
Gender × diets		0.300
Diets × age		0.047
Gender × age × diets		0.129

Note: Superscripts a, b, and c in the same column indicate a significant difference (*p* < 0.05).

**Table 6 animals-13-00335-t006:** The level of VFAs in the ruminal fluids of Tianzhu white yaks supplemented with various diets.

Month	Diet Treatment	Acetate (%)	Propionate (%)	Butyrate (%)	Isobutyrate (%)	Isovalerate (%)	Valerate (%)	TVFA (mmol/L)	A/P
6M	MSG (*n* = 3)	55.58 ± 1.45 ^a^	26.42 ± 0.39 ^a^	11.33 ± 0.10	2.11 ± 0.11 ^a^	2.37 ± 0.12 ^a^	2.18 ± 0.11 ^a^	27.28 ± 1.41 ^a^	2.10 ± 0.08 ^a^
	CSG (*n* = 3)	49.13 ± 0.77 ^b^	31.12 ± 0.07 ^b^	12.39 ± 0.06	2.35 ± 0.07 ^a^	2.56 ± 0.07 ^ab^	2.45 ± 0.09 ^a^	21.69 ± 1.11 ^b^	1.58 ± 0.06 ^b^
	G (*n* = 3)	45.49 ± 2.24 ^b^	33.55 ± 1.15 ^b^	12.48 ± 0.14	2.72 ± 0.11 ^b^	2.90 ± 0.13 ^b^	2.87 ± 0.10 ^b^	18.80 ± 0.90 ^b^	1.36 ± 0.11 ^b^
	*p*	0.012	0.012	0.710	0.016	0.041	0.005	0.006	0.003
30M	MSG (*n* = 3)	55.16 ± 0.84 ^a^	27.62 ± 1.00 ^a^	10.03 ± 0.95 ^a^	2.14 ± 0.13	2.84 ± 0.29	2.22 ± 0.07 ^a^	29.44 ± 4.40	2.00 ± 0.08 ^a^
	CSG (*n* = 3)	54.47 ± 1.38 ^a^	29.44 ± 0.83 ^ab^	9.52 ± 0.48 ^a^	2.15 ± 0.04	2.10 ± 0.09	2.31 ± 0.02 ^a^	21.37 ± 0.61	1.85 ± 0.09 ^a^
	G (*n* = 3)	43.16 ± 4.84 ^b^	32.41 ± 1.54 ^b^	15.97 ± 2.74 ^b^	2.42 ± 0.14	3.11 ± 0.68	2.89 ± 0.02 ^b^	23.90 ± 1.50	1.35 ± 0.20 ^b^
	*p*	0.049	0.049	0.066	0.195	0.300	0.020	0.180	0.038
54M	MSG (*n* = 3)	53.49 ± 0.54 ^a^	28.43 ± 1.78	13.63 ± 0.21 ^a^	1.93 ± 0.20	2.39 ± 0.14	2.06 ± 0.17 ^a^	25.87 ± 0.11 ^a^	1.89 ± 0.10
	CSG (*n* = 3)	49.86 ± 0.80 ^ab^	30.84 ± 0.55	10.37 ± 0.24 ^b^	2.40 ± 0.07	2.58 ± 0.01	2.54 ± 0.05 ^ab^	20.92 ± 0.12 ^b^	1.61 ± 0.03
	G (*n* = 3)	51.18 ± 0.83 ^b^	26.72 ± 1.71	12.24 ± 1.26 ^a^	2.40 ± 0.08	2.82 ± 0.01	2.72 ± 0.18 ^b^	21.30 ± 0.11 ^b^	1.93 ± 0.13
	*p*	0.035	0.217	0.023	0.078	0.148	0.049	< 0.001	0.125

Note: TVFA = total volatile fatty acids, A/P = acetate/propionate. Values in the same column with different lowercase superscript letters represent a significant difference at *p* < 0.05.

**Table 7 animals-13-00335-t007:** The VFA level in the ruminal fluids of Tianzhu white yaks of different ages.

Diet Treatment	Month	Acetate (%)	Propionate (%)	Butyrate (%)	Isobutyrate (%)	Isovalerate (%)	Valerate (%)	TVFA (mmol/L)	A/P
MSG	6M (*n* = 3)	55.58 ± 1.45	26.42 ± 0.39	11.33 ± 1.01 ^ab^	2.11 ± 0.11	2.37 ± 0.12	2.18 ± 0.06	27.28 ± 1.43	2.10 ± 0.08
	30M (*n* = 3)	55.16 ± 0.84	27.62 ± 1.00	10.03 ± 0.95 ^a^	2.14 ± 0.13	2.84 ± 0.29	2.22 ± 0.07	29.44 ± 4.40	2.00 ± 0.08
	54M (*n* = 3)	51.56 ± 0.54	28.43 ± 1.78	13.63 ± 0.21 ^b^	1.93 ± 0.20	2.391 ± 0.14	2.06 ± 0.17	25.87 ± 0.11	1.83 ± 0.10
	*p*	0.371	0.531	0.054	0.627	0.250	0.616	0.658	0.331
CSG	6M (*n* = 3)	49.13 ± 0.77 ^a^	31.12 ± 0.73	12.39 ± 0.60 ^a^	2.35 ± 0.07 ^ab^	2.56 ± 0.07 ^a^	2.45 ± 0.09	21.69 ± 0.90	1.58 ± 0.06 ^a^
	30M (*n* = 3)	54.47 ± 1.38 ^b^	29.44 ± 0.83	9.52 ± 0.48 ^b^	2.15 ± 0.04 ^a^	2.10 ± 0.09 ^b^	2.31 ± 0.22	21.37 ± 1.11	1.85 ± 0.09 ^b^
	54M (*n* = 3)	49.86 ± 0.80 ^a^	30.84 ± 0.55	10.37 ± 0.24 ^b^	2.40 ± 0.07 ^b^	2.58 ± 0.01 ^a^	2.54 ± 0.05	20.92 ± 0.12	1.61 ± 0.03 ^ab^
	*p*	0.020	0.282	0.013	0.075	0.004	0.126	0.769	0.060
G	6M (*n* = 3)	45.49 ± 2.24	33.55 ± 1.55 ^a^	12.48 ± 1.40	2.72 ± 0.11	2.90 ± 0.13	2.87 ± 0.10	18.80 ± 0.90 ^a^	1.36 ± 0.11
	30M (*n* = 3)	43.16 ± 4.84	32.41 ± 1.54 ^a^	15.97 ± 2.74	2.42 ± 0.13	3.11 ± 0.68	2.93 ± 0.22	23.90 ± 1.50 ^b^	1.38 ± 0.17
	54M (*n* = 3)	51.18 ± 0.83	26.72 ± 1.71 ^b^	12.24 ± 1.26	2.40 ± 0.08	2.82 ± 0.01	2.72 ± 0.18	21.30 ± 0.11 ^ab^	1.93 ± 0.13
	*p*	0.237	0.028	0.508	0.096	0.909	0.758	0.035	0.053

Note: TVFA = total VFAs, A/P = acetate/propionate. Values in the same column with different lowercase superscript letters represent a significant difference at *p* < 0.05.

**Table 8 animals-13-00335-t008:** Alpha diversity results of rumen microflora in Tianzhu white yaks supplemented with various diets.

Month	Indexes	MSG	CSG	G	*p*
6M	Shannon	5.27 ± 0.19	5.26 ± 0.06	5.01 ± 0.51	0.553
	Simpson	0.02 ± 0.00	0.02 ± 0.00	0.02 ± 0.01	0.750
	Ace	1306.62 ± 50.02	1323.93 ± 110.85	1167.95 ± 271.84	0.515
	Chao	1324.44 ± 33.49	1355.71 ± 133.66	1198.87 ± 284.54	0.569
30M	Shannon	5.30 ± 0.11	5.12 ± 0.37	4.78 ± 0.49	0.284
	Simpson	0.01 ± 0.00	0.03 ± 0.01	0.04 ± 0.04	0.432
	Ace	1383.76 ± 55.99 ^a^	1368.35 ± 129.77 ^a^	1093.15 ± 10.45 ^b^	0.008
	Chao	1409.77 ± 45.01 ^a^	1417.73 ± 116.85 ^a^	1094.16 ± 7.17 ^b^	0.002
54M	Shannon	5.19 ± 0.35	5.08 ± 0.03	5.10 ± 0.24	0.858
	Simpson	0.02 ± 0.01	0.02 ± 0.00	0.02 ± 0.00	0.933
	Ace	1287.73 ± 90.01	1254.94 ± 15.99	1279.90 ± 114.13	0.886
	Chao	1324.84 ± 91.54	1278.72 ± 44.80	1308.16 ± 116.34	0.820

Note: Values in the same row with different lowercase superscript letters represent a significant difference at *p* < 0.05.

**Table 9 animals-13-00335-t009:** Alpha diversity results of rumen microflora in Tianzhu white yaks of different ages.

Diet Treatment	Indexes	6M	30M	54M	*p*
MSG	Shannon	5.27 ± 0.18	5.29 ± 0.10	5.18 ± 0.35	0.850
	Simpson	0.01 ± 0.00	0.01 ± 0.00	0.02 ± 0.01	0.540
	Ace	1306.61 ± 50.01	1383.76 ± 55.99	1287.72 ± 99.00	0.261
	Chao	1324.44 ± 33.99	1409.76 ± 45.01	1324.83 ± 91.54	0.231
CSG	Shannon	5.25 ± 0.06	5.12 ± 0.37	5.08 ± 0.03	0.625
	Simpson	0.01 ± 0.00	0.02 ± 0.00	0.01 ± 0.00	0.104
	Ace	1323.92 ± 110.84	1368.34 ± 129.77	1254.94 ± 15.99	0.422
	Chao	1355.70 ± 133.66	1417.72 ± 116.84	1278.72 ± 44.79	0.339
G	Shannon	5.01 ± 0.51	4.78 ± 0.49	5.10 ± 0.24	0.666
	Simpson	0.02 ± 0.01	0.04 ± 0.04	0.02 ± 0.00	0.589
	Ace	1167.95 ± 271.84	1093.15 ± 10.45	1279.90 ± 114.13	0.450
	Chao	1198.87 ± 284.54	1094.16 ± 7.17	1308.16 ± 116.34	0.395

Note: Values in the same row with different lowercase superscript letters represent a significant difference at *p* < 0.05.

## Data Availability

The datasets presented in this study can be found in online repositories. The names of the repository/repositories and accession numbers can be found at https://submit.ncbi.nlm.nih.gov/subs/ PRJNA847264 (accessed on 1 September 2022).

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
