# Peer review of "Effects of Two Different Straw Pellets on Yak Growth Performance and Ruminal Microbiota during Cold Season"

_animals, 2023, doi:10.3390/ani13030335_

Round 1
Reviewer 1 Report (Previous Reviewer 1)
Paper adds to the limited, but growing, depth of knowledge regarding yak nutrition and production. Overall, I believe this area of research is very beneficial, at least on a largely regional level, as there are relatively few yaks outside of the Tibet Plateau.
While I appreciate the challenges of working with production first animals, the small sample sizes and only male yaks being used for rumen characteristics limits the scope of inference. Were forage samples taken of the grazed material? Do you have an indication of the nutrient content of the total diet of each treatment? Was the supplemental feed amount a consistent percentage of body weight?
In table 4, what p-value do the superscripts belong to? In Table 5, the diets p-value of 0.000 is not clear, how is it 0.000? Please double check references, especially #27 as it looks incomplete.
Author Response
To Reviewer:
We are very grateful for your assessment of our work. Thank you very much for your time and effort in reviewing our manuscript. We appreciate your useful comments. Please find our responses to the reviewer’s suggestions in the following section.
Reviewer #1 :
Q1: While I appreciate the challenges of working with production first animals, the small sample sizes and only male yaks being used for rumen characteristics limits the scope of inference. Were forage samples taken of the grazed material?
Response:
Thank you for carefully reading our manuscript and giving us valuable suggestion.
Unfortunately, we did not collect pasture samples because yaks in the grazing and supplementation groups were fed on the same pasture during the day and at the same time. Therefore, we did not consider the effect of pasture samples on this experiment, and we fixed the pasture samples on the same variable.
In addition, the description of pasture samples in the manuscript has been revised and added based on valuable comments from previous reviewers. We refer to the species of pasture grasses in Tianzhu pasture mentioned in the paper of Qi et al. “The pasture species in the Qinghai-Tibet Plateau at Tianzhu were diverse, and the major species included grasses (predominantly Elymus nutans), legumes (predorminantly Melissitus ruthenica), sedge (Kobresia humilis) and forbs (e.g. Potentilla bifurca, Gentiana straminea)”.
References:
Qi, J.; Nie, Z.; Jiao, T.; Zhang, D. Phosphorus and Defoliation Interact and Improve the Growth and Composition of the Plant Community and Soil Properties in an Alpine Pasture of Qinghai-Tibet Plateau. Plos One 2015, 10, e0141701.
Q2: Do you have an indication of the nutrient content of the total diet of each treatment?
Response:
Thank you for carefully reading our manuscript and giving us valuable suggestion.
In the nutrient content of the supplementary diet treatment group, we have measured crude protein, dry matter, neutral detergent fiber and acid detergent fiber. The details are as follows Table:
Table 1. Nutrient composition of roughage (after air drying).
|
Diet treatment |
Dry matter/ (%) |
Crude protein (%) |
Neutral detergent fiber (%) |
Acid detergent fiber (%) |
|
Corn straw |
91.57 |
3.30 |
63.55 |
41.98 |
|
Mixed straw |
94.76 |
7.09 |
58.78 |
38.79 |
Q3: Was the supplemental feed amount a consistent percentage of body weight?
Response:
Thank you for carefully reading our manuscript and giving us valuable suggestion.
The amount of feed supplementation is based on the percentage of the yak's body weight.
Q4: In table 4, what p-value do the superscripts belong to?
Response:
Thank you for carefully reading our manuscript and giving us valuable suggestion.
The p-value indicates the significance of gender for the average daily weight gain of yaks at the same age and supplementary feeding conditions.
Q5: In Table 5, the diets p-value of 0.000 is not clear, how is it 0.000?
Response:
Thank you for carefully reading our manuscript and giving us valuable suggestion.
We have revised P= 0.000 to P < 0.001 in manuscript.
Q6: Please double check references, especially #27 as it looks incomplete.
Response:
Thank you for carefully reading our manuscript and giving us valuable suggestion.
We have made correction according to your comments in manuscript.
References:
[27] Magoc, T.; Salzberg, S.L. FLASH: fast length adjustment of short reads to improve genome assemblies. Bioinformatics 2011, 27, 2957-2963.
Special thanks to you for your good comments.
We tried our best to improve the manuscript and made some changes in the manuscript. These changes will not influence the content and framework of the paper. And here we did not list the changes but marked in red in revised paper.
We appreciate for Editors/Reviewers' warm work earnestly, and hope that the correction will meet with approval.
Once again, thank you very much for your comments and suggestions.
Reviewer 2 Report (Previous Reviewer 2)
I have no more suggestions.
Author Response
We are very grateful for your assessment of our work. Thank you very much for your time and effort in reviewing our manuscript.
We appreciate for Reviewers' warm work earnestly, and hope that the correction will meet with approval.
Round 2
Reviewer 1 Report (Previous Reviewer 1)
No more at this time.
This manuscript is a resubmission of an earlier submission. The following is a list of the peer review reports and author responses from that submission.
Round 1
Reviewer 1 Report
Review for Manuscript ID: animals-2038490
Summary: This article discusses the supplementation of Tianzhu white yaks during the winter with different straw-based pellets. This paper discusses not only animal growth, but also rumen fermentation and microbiome. Considering the importance of these animals to the region, this paper seems to address a topic of need for increasing animal productivity and the addition of the microbiome data increases the scientific appeal.
General comments about the article: While I believe this research has strong benefit to yak producers in China / Tibet, there seems to be scientific missteps taken during the project. Having 116 animals is a strong number of experimental units, however that number is divided so much that the end result is only a few animals per treatment, which would limit the statistical power of the research. Why were only 3 males of the 3 age groups harvested and utilized for rumen data? What was the rationale for using animals of only the 6, 30, and 54-month age groups? Do you believe the rumen microbiome / fermentation data would change from beginning of supplementation to end of supplementation? Was harvesting the animals at the end of the feeding period the only mechanism looked at for rumen fluid collection? Previous cattle research has utilized oral lavage (Lodge-Ivey et al., 2009) as a means to collect rumen samples from multiple live animals. Were the male yaks intact bulls and if so could this impact growth and / or rumen fermentation? In the discussion, it is mentioned that in the 6 month old yaks, the rumen and flora are immature; what evidence (either results or previous literature) would support this claim? To me, your data shows about as much diversity in the 6 month old yaks as the older yaks.
General comments about the review: The review covers the topic fairly well, given the somewhat limited research on yaks as a whole. The authors cover a number of comparable species which should have some relation to the yak as a whole. There are likely some other papers which could relate and some referenced papers have fairly limited relation to the research objective of the project (# 20 discussed in the Introduction for example).
Specific comments: Line 53 & 54, is the 0.7% of worldwide yak population truly 100,000? Please confirm with the reference that these numbers are correct, as I was under the impression that there are many fewer than 1.5 million yaks worldwide. Line 158 – 166, is there a reference for this procedure or are you the first to conduct this analysis in this manner? Similarly, for Lines 167 – 178 is the reference # 25 just for the primers or for the process as a whole? For Lines 179 – 194, is there a reference for this procedure? For Tables 2 & 3, add “c” to the superscripts definition in the legend at the bottom of each table. Also for Table 3, in the 30 M – MSG female group, it should be “0.20” instead of “20”. Lines 358 – 365 is confusing and does not portray a clear picture of your results. Also, wouldn’t a change in microbial population and therefore fermentation parameter better explain the change in propionate as opposed to energy deficiency and gluconeogenesis, as the animal would likely use the newly made glucose for it’s own energy? I am also not sure if this reference (# 32) is fully supportive of your description, please double-check and confirm.
Author Response
To Reviewer:
We are very grateful for your assessment of our work. Thank you very much for your time and effort in reviewing our manuscript. We appreciate your useful comments. Please find our responses to the reviewer’s suggestions in the following section.
Q1: General comments about the article: While I believe this research has strong benefit to yak producers in China / Tibet, there seems to be scientific missteps taken during the project. Having 116 animals is a strong number of experimental units, however that number is divided so much that the end result is only a few animals per treatment, which would limit the statistical power of the research. Why were only 3 males of the 3 age groups harvested and utilized for rumen data?
Response:
Thank you for carefully reading our manuscript and giving us valuable suggestion.
The 116 Tianzhu white yaks were only selected and used by breeding test animals, while only 27 Tianzhu white yaks were finally slaughtered. The only three yaks were selected as research yaks in three age groups for several reasons, first of all, because there are fewer yaks to choose from in the same pasture and the same breeding level, so under the restrictions of the same pasture conditions, in order to test the neatness and uniformity of the test samples, we slaughter according to the minimum number of replicates of the samples contained in the group(n=3). Secondly, we only chose male yaks as the research object, mainly because male yaks are mainly slaughtered for meat product processing, while female yaks are generally retained for reproduction
Q2: What was the rationale for using animals of only the 6, 30, and 54-month age groups?
Response:
Thank you for carefully reading our manuscript and giving us valuable suggestion.
We selected 6-month-old, 30-month-old and 54-month-old yaks as the study subjects, because 6 months old is the yak calving stage, 30 months old is the breeding stage of yaks, and 54 months old is the adult stage of the yak, explored the changes of daily weight gain and rumen microorganisms of yaks may be more representative of straw feed supplementation through these three periods. Wang et al. analyzed the role of tsk 4 in testicular development, they also selected 6 months of age as pre-sexual maturity and 30 months of age as sexual maturity in yaks [1]. Xu et al in identifying the role of LncRNA H19 in muscle growth and development of yaks, the 6-month-old calves and 36-month-old adult cattle were selected for the study [2]. In addition, our previous study based on yak calves (6 months old), young adults (30 months old) and adults (54 months old) to explore their differentially expressed genes, and we found significant differences in muscle fiber diameter occurred were observed at these three ages [3].
References:
[1] Wang, X.; Pei, J.; Xiong, L.; Guo, S.; Cao, M.; Kang, Y.; Bao, P.; Wu, X.; Chu, M.; Liang, C.; et al. Identification of the TSSK4 Alternative Spliceosomes and Analysis of the Function of the TSSK4 Protein in Yak (Bos grunniens). Animals (Basel) 2022, 12,
[2] Xu, X.; Ji, S.; Li, W.; Yi, B.; Li, H.; Zhang, H.; Ma, W. LncRNA H19 promotes the differentiation of bovine skeletal muscle satellite cells by suppressing Sirt1/FoxO1. Cell Mol Biol Lett 2017, 22, 10
[3] Shi, B.; Shi, X.; Zuo, Z.; Zhao, S.; Zhao, Z.; Wang, J.; Zhou, H.; Luo, Y.; Hu, J.; Hickford, J.G.H. Identification of differentially expressed genes at different post-natal development stages of longissimus dorsi muscle in Tianzhu white yak. Gene 2022, 823, 146356
Q3: Do you believe the rumen microbiome/fermentation data would change from beginning of supplementation to end of supplementation?
Response:
Thank you for carefully reading our manuscript and giving us valuable suggestion.
Species, diet, age, and season influence rumen microbes and rumen fermentation, of which diet is a key factor [1]. The supplementation of concentrate to the diet of ruminants changed the fermentation of rumen microorganisms and increased the abundance of some core rumen microorganisms [2]. In this study, we found that the rumen content of volatile fatty acids, such as total acid and acetic acid was varied in yaks fed corn straw and mixed straw diets at 6 M, 30 M and 54 months compared to the grazing group. In addition, we found that the abundance of rumen microorganisms was also changed, such as the abundance of Christensenellaceae_R-7_group was significantly increased after supplementation with straw feed. Finally, the results of the Beta diversity showed that yaks in the supplemented (corn straw and mixed straw) and grazed groups formed different rumen microflora. Thus, we consider the rumen microbiome and fermentation data have changed from beginning of supplementation to end of supplementation.
References:
[1] Henderson, G.; Cox, F.; Ganesh, S.; Jonker, A.; Young, W.; Global Rumen Census, C.; Janssen, P.H. Rumen microbial community composition varies with diet and host, but a core microbiome is found across a wide geographical range. Sci Rep 2015, 5, 14567
[2] Liu, H.; Xu, T.; Xu, S.; Ma, L.; Han, X.; Wang, X.; Zhang, X.; Hu, L.; Zhao, N.; Chen, Y.; et al. Effect of dietary concentrate to forage ratio on growth performance, rumen fermentation and bacterial diversity of Tibetan sheep under barn feeding on the Qinghai-Tibetan plateau. PeerJ 2019, 7, e7462
Q4: Was harvesting the animals at the end of the feeding period the only mechanism looked at for rumen fluid collection? Previous cattle research has utilized oral lavage (Lodge-Ivey et al., 2009) as a means to collect rumen samples from multiple live animals.
Response:
Thank you for carefully reading our manuscript and giving us valuable suggestion.
Ruminal fluid can be collected by surgical fistula collection and oral lavage. Surgical fistula collection and oral irrigation are generally used to collect rumen fluid mainly in live animals We did not adopt the fistula technique and oral irrigation mainly because we also measured the changes in meat quality and traits of the test yaks which required slaughter of the yaks so we did not consider fistula and oral irrigation for rumen fluid collection, and secondly we learned that the fistula technique can disrupt the anaerobic environment of the rumen and lead to differences in rumen microbial function and diversity [1].
References:
[1] Cao, Y.; Zhu, B.; Li, F.; Zhang, D.; Guo, T.; Li, F.; Yang, G. The Destruction of the Anaerobic Environment Caused by Rumen Fistula Surgery Leads to Differences in the Rumen Microbial Diversity and Function of Sheep. Front Vet Sci 2021, 8, 754195
Q5: Were the male yaks intact bulls and if so could this impact growth and / or rumen fermentation?
Response:
Thank you for carefully reading our manuscript and giving us valuable suggestion.
We used intact yaks, this experimental animal did not involve castrated yaks, and this study only considered the effects of supplementary feeding on yak rumen growth performance and rumen fermentation. In confined feeding operations, bulls outweigh intact counterparts in growth performance [1-3], The effects of castration on yak rumen fermentation require further study.
References:
[1] Seideman S.C., Cross H.R., Oltjen R.R., Schanbacher B.D. Utilization of the intact male for red meat production: A review. J. Anim. Sci. 1982;55:826–840
[2] Venkata Reddy B., Sivakumar A.S., Jeong D.W., Woo Y.-B., Park S.J., Lee S.Y., Byun J.-Y., Kim C.H., Cho S.H., Hwang I. Beef quality traits of heifer in comparison with steer, bull, and cow at various feeding environments. Anim. Sci. J. 2014;86:11–16
[3] Ismail H.A. Master’s Thesis. University of Khartoun; Khartoun State, Sudan: 2006. Effect of Castration on Feedlot Performance, Carcass Characteristics and Meat Quality of Western Sudan Baggara Bulls.
Q6: In the discussion, it is mentioned that in the 6 months old yaks, the rumen and flora are immature; what evidence (either results or previous literature) would support this claim? To me, your data shows about as much diversity in the 6 months old yaks as the older yaks.
Response:
Thank you for carefully reading our manuscript and giving us valuable suggestion.
Jami, Israel et al. reported significantly different composition of rumen microbes from birth to 2 years of age in cattle [1]. Also, the rumen microbial ecology of cattle developed further from the intake of solid feeds [2], while 6-month-old yaks were transformed from liquid to solid feeds. So we speculate that 6-month-old yaks are gradually maturing and stabilizing. lachnospiraceae_NK3A20_group plays an important role in maintaining intestinal health. In our study we found the highest relative abundance of lachnospiraceae_NK3A20_group in 6-month-old yaks compared to other months of age. So we speculate that this may be due to the immature development of rumen flora in 6-month-old yaks. As you mentioned there was no significant difference Alpha diversity of rumen microflora between 6 and 54 months of age, but this does notfully indicate that there is no difference in rumen microflora between 6 and 54 months of age. Forther more, in our study beta diversity of rumen microbiota results showed that rumen microbial diversity in 6M yaks was significantly different from 30M and 54M yaks.
References:
[1] Jami E, Israel A, Kotser A, Mizrahi I. Exploring the bovine rumen bacterial community from birth to adulthood. ISME J. 2013 Jun;7(6):1069-79.
[2] Dill-McFarland, K.A.; Breaker, J.D.; Suen, G. Microbial succession in the gastrointestinal tract of dairy cows from 2 weeks to first lactation. Sci Rep 2017, 7, 40864
Q7: General comments about the review: The review covers the topic fairly well, given the somewhat limited research on yaks as a whole. The authors cover a number of comparable species which should have some relation to the yak as a whole. There are likely some other papers which could relate and some referenced papers have fairly limited relation to the research objective of the project (# 20 discussed in the Introduction for example).
Response:
Thank you for carefully reading our manuscript and giving us valuable suggestion.
We list the effects of straw feed on the content of unsaturated fatty acids in goat meat quality mainly to illustrate the advantages of straw feed for ruminants, which can not only improve the daily weight gain of ruminants, improve the environment in the rumen, but also improve their meat quality traits.
Q8: Line 53 & 54, is the 0.7% of worldwide yak population truly 100,000? Please confirm with the reference that these numbers are correct, as I was under the impression that there are many fewer than 1.5 million yaks worldwide.
Response:
Thank you for carefully reading our manuscript and giving us valuable suggestion.
We have recalculated that the Tianzhu white yak accounts for 7 % of the world's yaks.
References:
[1] Zhang, X.; Wang, K.; Wang, L.; Yang, Y.; Ni, Z.; Xie, X.; Shao, X.; Han, J.; Wan, D.; Qiu, Q. Genome-wide patterns of copy number variation in the Chinese yak genome. BMC Genomics 2016, 17, 379
Q9: Line 158 – 166, is there a reference for this procedure or are you the first to conduct this analysis in this manner?
Response:
Thank you for carefully reading our manuscript and giving us valuable suggestion.
Determination of volatile fatty acids according to the method of Zhang et al [1]. We have added relevant references in the manuscript.
References:
[1] Zhang, Q.; Degen, A.; Hao, L.; Huang, Y.; Niu, J.; Wang, X.; Chai, S.; Liu, S. An increase in dietary lipid content from different forms of double-low rapeseed reduces enteric methane emission in Datong yaks on the Qinghai-Tibetan Plateau. Anim Sci J 2020, 91, e13489
Q10: Lines 167 – 178 is the reference # 25 just for the primers or for the process as a whole?
Response:
Thank you for carefully reading our manuscript and giving us valuable suggestion.
The 25th reference was the reference used for the primer.
Q11: For Lines 179 – 194, is there a reference for this procedure?
Response:
Thank you for carefully reading our manuscript and giving us valuable suggestion.
Based on the raw data obtained from the Illumina HiSeq (2500 PE250) sequencing platform, original sequence quality was controlled and spliced using the Fastp software perform (https://ccb.jhu.edu/software/FLASH/index.shtml) [1]. The sequences were clustered to operational taxonomic units (units OTU) using UPARSE software performs (http://drive5.com/uparse/) based on 97% similarity [2]. The Uchime algorithm was used to eliminate chimeras by Usearch software. Each sequence was classified and annotated at different levels (phylum, genus) through the Ribosomal Database Project (RDP) [3], and then the alignment database (SSU123) was then compared with the entries in the Silva database [4]. Alpha diversity (Shannon and Simpson) and richness (Ace and Chao) were analyzed using mother software platform (https://www.mothur.org/wiki/Downloadmothur) [5].
We have supplemented the relevant references in manuscript.
References:
[1] agoc, T.; Salzberg, S.L. FLASH: fast length adjustment of short reads to improve genome assemblies. Bioinformatics 2011, 27, 2957-2963
[2] Edgar, R.C. UPARSE: highly accurate OTU sequences from microbial amplicon reads. Nat Methods 2013, 10, 996-998
[3] Cole, J.R.; Wang, Q.; Cardenas, E.; Fish, J.; Chai, B.; Farris, R.J.; Kulam-Syed-Mohideen, A.S.; McGarrell, D.M.; Marsh, T.; Garrity, G.M.; et al. The Ribosomal Database Project: improved alignments and new tools for rRNA analysis. Nucleic Acids Res 2009, 37, D141-145
[4] Quast, C.; Pruesse, E.; Yilmaz, P.; Gerken, J.; Schweer, T.; Yarza, P.; Peplies, J.; Glockner, F.O. The SILVA ribosomal RNA gene database project: improved data processing and web-based tools. Nucleic Acids Res 2013, 41, D590-596
[5] Wang, Y.; Sheng, H.F.; He, Y.; Wu, J.Y.; Jiang, Y.X.; Tam, N.F.; Zhou, H.W. Comparison of the levels of bacterial diversity in freshwater, intertidal wetland, and marine sediments by using millions of illumina tags. Appl Environ Microbiol 2012, 78, 8264-8271
Q12: For Tables 2 & 3, add “c” to the superscripts definition in the legend at the bottom of each table. Also for Table 3, in the 30 M – MSG female group, it should be “0.20” instead of “20”.
Response:
Thank you for carefully reading our manuscript and giving us valuable suggestion.
We have made correction according to your comments in manuscript.
Q13: Lines 358 – 365 is confusing and does not portray a clear picture of your results. Also, wouldn’t a change in microbial population and therefore fermentation parameter better explain the change in propionate as opposed to energy deficiency and gluconeogenesis, as the animal would likely use the newly made glucose for it’s own energy? I am also not sure if this reference (# 32) is fully supportive of your description, please double-check and confirm.
Response:
Thank you for carefully reading our manuscript and giving us valuable suggestion.
There are various factors that cause changes in propionic acid content in animals. Changes in propionic acid concentrations in ruminants may be attributed to rumen microbial abundance and the composition of diets [1-2]. Propionic acid, as the main substrate of gluconeogenesis, provides energy to the animal organism through the gluconeogenesis pathway [3]. We found high levels of propionic acid in the CSG groups and G groups, which confirms our speculation that yaks in the CSG groups and G groups dietary energy insufficiency cannot meet their own energy needs and therefore need higher concentrations of propionic acid to promote more glucose to provide energy. Secondly, the correlation analysis found that the Christensenellaceae_R-7_group and NK4A214_group abundances were negatively correlated with the propionic acid content [4], and the Christensenellaceae_R-7_group and NK4A214_group abundances in the CSG group and G group were lower than those in the MSG group, which was consistent with the change of propionic acid concentration caused by the change of body energy.
References:
[1] Serment, A.; Schmidely, P.; Giger-Reverdin, S.; Chapoutot, P.; Sauvant, D. Effects of the percentage of concentrate on rumen fermentation, nutrient digestibility, plasma metabolites, and milk composition in mid-lactation goats. J Dairy Sci 2011, 94, 3960-3972
[2] De Vadder, F.; Kovatcheva-Datchary, P.; Zitoun, C.; Duchampt, A.; Backhed, F.; Mithieux, G. Microbiota-Produced Succinate Improves Glucose Homeostasis via Intestinal Gluconeogenesis. Cell Metab 2016, 24, 151-157
[3] Saro, C.; Mateo, J.; Caro, I.; Carballo, D.E.; Fernandez, M.; Valdes, C.; Bodas, R.; Giraldez, F.J. Effect of Dietary Crude Protein on Animal Performance, Blood Biochemistry Profile, Ruminal Fermentation Parameters and Carcass and Meat Quality of Heavy Fattening Assaf Lambs. Animals (Basel) 2020, 10,
[4] Wang, Y.; Shen, Q.; Zhong, S.; Chen, Y.; Yang, Y. Comparison of Rumen Microbiota and Serum Biochemical Indices in White Cashmere Goats Fed Ensiled or Sun-Dried Mulberry Leaves. Microorganisms 2020, 8,
Special thanks to you for your good comments.
We tried our best to improve the manuscript and made some changes in the manuscript. These changes will not influence the content and framework of the paper. And here we did not list the changes but marked in red in revised paper.
We appreciate for Editors/Reviewers' warm work earnestly, and hope that the correction will meet with approval.
Once again, thank you very much for your comments and suggestions.
Reviewer 2 Report
The manuscript reported the results of supplementing different straw pellets on growth performance and ruminal microbiota of grazing yaks. The results are practically useful to yak farming. The results were clearly presented. However, the mansucript need to be revised in the following aspects.
1. The environmental conditions including ambient temperature, pasture, altitude etc. for yaks, need to be described.
2. Othe rumen fermentation parameters except VFA should be measured or analysed and presented in the manuscript, including ruminal pH, ruminal concentrations of ammonia nitrogen and microbial crude protein.
3. Line 125: The composition of the premix needs to be described in detail.
4. Lines 125-126 and Lines 142-143: repeated description.
5. Tables 3-6: How many yaks were used for each treatment group? Need to indicate.
6. The intakes of the straw pellets and pasture of the yaks need to be provided in Results section.
7. I agree ruminal microbiota are important to explain the growth rate of the yaks. But I would suggest discuss the weight gain of the yaks based on the actual nutrient intakes (mainly crude protein and energy) and the nutrient requirements (mainly crude protein and energy).
Author Response
To Reviewer:
We are very grateful for your assessment of our work. Thank you very much for your time and effort in reviewing our manuscript. We appreciate your useful comments. Please find our responses to the reviewer’s suggestions in the following section.
Q1: The environmental conditions including ambient temperature, pasture, altitude etc. for yaks, need to be described.
Response:
Thank you for carefully reading our manuscript and giving us valuable suggestion.
We have added the elevation and latitude of the yak rearing pastures in Line 122, Pasture conditions have added in Line 133.
Q2: Other rumen fermentation parameters except VFA should be measured or analysed and presented in the manuscript, including ruminal pH, ruminal concentrations of ammonia nitrogen and microbial crude protein.
Response:
Thank you for carefully reading our manuscript and giving us valuable suggestion.
I'm sorry, we are publishing data on pH and ruminal concentrations of ammonia nitrogen and microbial crude protein in other journals.
Q3: Line 125, The composition of the premix needs to be described in detail.
Response:
Thank you for carefully reading our manuscript and giving us valuable suggestion.
The premix recipe is shown in table, which we have supplemented in the manuscript in the form of an attachment. We have supplemented in the Line 130.
The composition of the premix.
|
Item |
VA |
VD3 |
VE |
Ca |
P |
Cu |
Fe |
Zn |
|
content/(kg) |
90 000 IU |
30 000 IU |
350 IU |
120 g |
20 g |
160 mg |
1200 mg |
2000 mg |
Q4: Lines 125-126 and Lines 142-143: repeated description.
Response:
Thank you for carefully reading our manuscript and giving us valuable suggestion.
We have deleted the repeated about “The mixed straw and corn straw pellets (3–5 cm) were made using in length by a straw cutter” in the Line 130.
Q5: Tables 3-6: How many yaks were used for each treatment group? Need to indicate.
Response:
Thank you for carefully reading our manuscript and giving us valuable suggestion.
We have added the number of yaks for each treatment group in Table3-6 according to the Reviewer's comments.
Q6: The intakes of the straw pellets and pasture of the yaks need to be provided in Results section.
Response:
Thank you for carefully reading our manuscript and giving us valuable suggestion.
Regarding the average daily feed at different age we mentioned in the material methods, “each yak in the MSG groups and CSG groups at 6M, 18M, 30M, 42M and 54M was fed 1, 2, 2.5, 3 and 3.5 kg of mixed straw pellets and corn straw pellets at 6:pm respectively every day.” The actual daily feed intake of each group of yaks at different ages is shown in the table. Yaks feed freely during grazing, so it is impossible to calculate their feed intake.
|
Gender |
Diet treatment |
Month |
|||||
|
6M |
18M |
30M |
42M |
54M |
|||
|
male |
MSG |
1.01±0.02 |
1.95±0.02 |
2.45±0.05 |
2.99±0.00 |
3.43±0.02 |
|
|
CSG |
0.92±0.02 |
1.97±.0.02 |
2.43±0.08 |
2.80±0.06 |
3.48±0.03 |
||
|
female |
MSG |
1.00±0.02 |
1.94±0.01 |
2.43±0.02 |
2.89±0.03 |
3.32±0.04 |
|
|
CSG |
1.02±.0.02 |
1.97±0.01 |
2.47±0.00 |
2.80±0.06 |
3.36±0.03 |
|
|
Q7: I agree ruminal microbiota are important to explain the growth rate of the yaks. But I would suggest discuss the weight gain of the yaks based on the actual nutrient intakes (mainly crude protein and energy) and the nutrient requirements (mainly crude protein and energy).
Response:
Thank you for carefully reading our manuscript and giving us valuable suggestion.
Based on your recommendations, we supplemented the effect of dietary energy levels on yak growth performance in manuscript.
“Dietary energy utilization is the main factor affecting the performance of livestock, and the growth efficiency of livestock mainly depends on the energy level of the diet. The MSG exhibited a better growth performance than the CSG group, which can be attributed to the higher energy in the MSG group than those in the CSG group. This result is consistent with previous studies on yaks supplemented with high energy diets and medium energy diets compared with low energy diets, who exhibited better growth performance [1]. Thus, the MSG groups provided more energy than the CSG groups, which was may reason why the growth performance of the MSG groups was higher than that of CSG groups.”
References:
[1] Wang, H.; Li, H.; Wu, F.; Qiu, X.; Yu, Z.; Niu, W.; He, Y.; Su, H.; Cao, B. Effects of Dietary Energy on Growth Performance, Rumen Fermentation and Bacterial Community, and Meat Quality of Holstein-Friesians Bulls Slaughtered at Different Ages. Animals (Basel) 2019, 9.
Special thanks to you for your good comments.
We tried our best to improve the manuscript and made some changes in the manuscript. These changes will not influence the content and framework of the paper. And here we did not list the changes but marked in red in revised paper.
We appreciate for Editors/Reviewers' warm work earnestly, and hope that the correction will meet with approval.
Once again, thank you very much for your comments and suggestions.
Round 2
Reviewer 1 Report
- I would suggest adding some more information as to the selection criteria of harvested animals and rumen fluid collection. Add some of the information given in the response to reviewer.
- I still believe that the discussion around propionate (lines 362- 370ish) is less about energy deficiency and more about microbe diversity and changes which result in different microbial populations, but that may just be a philosophical difference.
Author Response
We are very grateful for your assessment of our work. Thank you very much for your time and effort in reviewing our manuscript. We appreciate your useful comments. Please find our responses to the reviewer’s suggestions in the following section.
Q1: I would suggest adding some more information as to the selection criteria of harvested animals and rumen fluid collection. Add some of the information given in the response to reviewer.
Response:
Thank you for carefully reading our manuscript and giving us valuable suggestion.
We have provided the criteria of rumen fluid collection shown below.
Collection rumen fluid
Accuracy of sampling: Normal tissue samples should not contain diseased tissues. As far as conditions permit, we strive to keep the samples of the experimental and control groups as consistent as possible in terms of sampling time, site, and processing conditions. Data on various characteristics of representative samples must be accurately recorded and collected, prepared, stored, and transported for experimental processing as required.
Repeatability of sampling: Sampling of biological replicates should minimize the variation of samples between replicates. As far as possible, duplicate samples should be as consistent as conditions permit in terms of sampling time, site and processing conditions; otherwise, the reproducibility and credibility of experimental results may be affected.
Timeliness of sampling: Sample quality is the most critical factor affecting experimental results, so experimental samples used for research are collected, stored, transported and prepared as quickly as possible to minimize the time from sample collection to experimentation.
Low temperature of sampling: After the samples taken are isolated and processed by quick washing and labeling, they should be immediately put into liquid nitrogen for at least 3h and then stored in -80℃ refrigerator or dry ice to ensure that the samples are always at -80℃ before the experimental operation to avoid the degradation of RNA.
Specific steps for rumen contents sample collection
- After the death of the subject, the entire rumen is removed with a sterile scalpel, in a sterile state, and the contents of the desired rumen are cut out (on a sterile operating table, if conditions permit).
- The contents were scooped with a sterile scalpel, filtered through four layers of gauze and immediately placed on ice for dispensing and labeling.
- Dispensing in sterilized centrifuge tubes, single sample sampling volume: 200-500 mg/tube, to ensure the smooth conduct of the experiment, take multiple backup tubes of each sample and collect as many samples as sampling allows.
- After dispensing, if DNA is extracted on the same day, it can be stored at 4°C. For long-term storage, it should be put into liquid nitrogen quickly and then stored at -80°C immediately afterwards.
Q2: I still believe that the discussion around propionate (lines 362- 370) is less about energy deficiency
and more about microbe diversity and changes which result in different microbial populations, but that may just be a philosophical difference.
Response:
Thank you for carefully reading our manuscript and giving us valuable suggestion.
We have revised this section in manuscript according to the reviewer's request, as follows.
“Microbial abundance and diversity can cause changes in propionic acid content, such as Propionibacterium andAcidipropionibacterium abundance was correlated with propionate content [1]. In our study we observed a significant decrease in propionate concentration in the MSG group compared to group G. Similarly, our correlation results show that Christensenellaceae_R7_group and Ruminococcaceae_NK4A214_group were inversely correlated with propionate content. Therefore, the decrease in propionate content of the MSG group may be due to microbial composition and abundance.”
[1] Colomban, A.; Roger, L.; Boyaval, P. Production of propionic acid from whey permeate by sequential fermentation, ultrafiltration, and cell recycling. Biotechnol Bioeng 1993, 42, 1091-1098
Special thanks to you for your good comments.
We tried our best to improve the manuscript and made some changes in the manuscript. These changes will not influence the content and framework of the paper. And here we did not list the changes but marked in red in revised paper.
We appreciate for Editors/Reviewers' warm work earnestly, and hope that the correction will meet with approval.
Once again, thank you very much for your comments and suggestions.
Reviewer 2 Report
The authors revsied the manuscript according to most of the reviewers' comments. However, the results of ruminal pH and ammonia-nitrogen concentration were not supplied as required. The results in Tables 3 and 4, and the results in Tables 5 and 6 are also repeatedly presented. The interactions among gender, diet and age (months) were not statistically analyzed.
Author Response
We are very grateful for your assessment of our work. Thank you very much for your time and effort in reviewing our manuscript. We appreciate your useful comments. Please find our responses to the reviewer’s suggestions in the following section.
Q1:The authors revsied the manuscript according to most of the reviewers' comments. However, the results of ruminal pH and ammonianitrogen concentration were not supplied as required. The results in Tables 3 and 4, and the results in Tables 5 and 6 are also repeatedly presented. The interactions among gender, diet and age (months) were not statistically analyzed.
Response:
Thank you for carefully reading our manuscript and giving us valuable suggestion.
I'm sorry, because this data is being published in other journals, we cannot present it in the manuscript yet, so we present this data in the Table1 below. In addition, we determined that the pH range was between 6.2 and 6.6, which is the normal range of pH in the rumen environment of ruminants, so we did not damage the health of the rumen environment after supplementing with straw feed. In addition, the content of ammonia nitrogen and microorganisms was between 10-13. The high ammonia nitrogen content in the MSG group may be due to the high crude protein content in the diet, which increases rumen microbial activity and accelerates the degradation of nitrogenous substances in the rumen and the synthesis of NH3-N and MCP, and the increased synthesis of MCP is more beneficial to yak fattening.
In addition, Table 3 in the manuscript shows the change in the average daily weight gain (ADG) of Tianzhu white yak at different ages under the same supplementary feed and gender. Table 4 in the manuscript showed effect of gender on the average daily weight gain of the white yak under the same complementary food and age. Also, the data in Table 5 showed the effect of supplemental feeding on volatile fatty acids in yak rumen at the same age, the data in Table 6 showed the effect of ages on volatile fatty acids in yak rumen with the same supplementary feeding.
Finally, we have provided information on the effects of age and sex and diet interactions on ADG of yaks (Table 2). Pearson correlation analysis was performed using SPSS 26.0 bivariate simple correlation and two-tailed t-test for significance of differences, the results showed that age and diet were significantly correlated with daily weight gain, respectively.
Table 1 Effect of supplemental feeding on rumen pH and NH3-N in yaks.
|
Month |
Diet treatment |
pH |
NH3-N(mg/dL) |
MCP |
|
6M |
MSG (n=3) |
6.47±0.16 |
13.02±0.39a |
1.92±0.35a |
|
CSG (n=3) |
6.50±0.17 |
11.12±0.07b |
1.78±0.63b |
|
|
G (n=3) |
6.57±0.12 |
10.55±1.15b |
1.54±0.15c |
|
|
P |
0.078 |
0.032 |
0.032 |
|
|
30M |
MSG (n=3) |
6.18±0.84 |
12.82±0.43 |
1.97±0.56a |
|
CSG (n=3) |
6.24±0.14 |
12.12±0.07 |
1.78±0.54b |
|
|
G (n=3) |
6.27±0.19 |
12.03±0.15 |
1.77±0.25b |
|
|
P |
0.079 |
0.063 |
0.035 |
|
|
54M |
MSG (n=3) |
6.37±0.84 |
12.72±0.69 |
1.72±0.55 |
|
CSG (n=3) |
6.58±0.17 |
11.32±0.27 |
1.68±0.43 |
|
|
G (n=3) |
6.47±0.15 |
11.55±1.15 |
1.64±0.19 |
|
|
P |
0. 084 |
0.068 |
0.870 |
Table 2 the effects of age and sex and diet interactions on ADG of yaks
|
Item |
gender |
age |
diets |
ADG |
|
gender |
1 |
|
|
|
|
age |
0.11 |
1 |
|
|
|
diets |
-0.06 |
0.001 |
1 |
|
|
ADG |
-0.143 |
0.320** |
0.768** |
1 |
** represents significant (two-tailed) correlation at the 0.01 level (two-tailed)
Special thanks to you for your good comments.
We tried our best to improve the manuscript and made some changes in the manuscript. These changes will not influence the content and framework of the paper. And here we did not list the changes but marked in red in revised paper.
We appreciate for Editors/Reviewers' warm work earnestly, and hope that the correction will meet with approval.
Once again, thank you very much for your comments and suggestions.